# Methods of Urea Fertilizer Application Influence Growth, Yield, and Nitrogen Use Efficiency of Transplanted Aman Rice

Alpina Akter [1], Mohammad Rafiqul Islam [1,*], M. Rafiqul Islam [1], Md. Ahidul Islam [2], Samia Lutfa Hasan [1], Shihab Uddin [1] and Mohammad Mahmudur Rahman [3,4]

[1] Department of Soil Science, Bangladesh Agricultural University, Mymensingh 2202, Bangladesh
[2] Department of Animal Science, Patuakhali Science and Technology University, Dumki, Patuakhali 8602, Bangladesh
[3] Global Centre for Environmental Remediation (GCER), College of Engineering, Science and Environment, The University of Newcastle, Callaghan 2308, Australia
[4] Department of General Educational Development, Faculty of Science & Information Technology, Daffodil International University, Ashulia, Savar, Dhaka 1207, Bangladesh
* Correspondence: rafiqss69@bau.edu.bd

**Abstract:** Although urea placed deep reduces nitrogen (N) loss and increases rice yield, its use is not expanding due to the lack of effective application methods. A study was carried out to determine how different urea application techniques affected the yield and nitrogen use efficiency (NUE) of transplanted Aman rice (cv. BRRI dhan46). The experiment was set up in a RCBD design with seven treatments: $T_1$ (deep placement of urea briquettes (DPUB) by hand), $T_2$ (DPUB by battery-powered applicator), $T_3$ (deep placement of prilled urea (PU) briquettes by BRRI applicator), $T_4$ (DPUB by injector applicator), $T_5$ (DPUB by push-type applicator), $T_6$ (broadcast application of PU), $T_7$ (zero-N), and three replications. Findings showed that the $NH_4^+$-N concentration in field water peaked on day 3 and then rapidly fell as time passed, while the $NO_3^-$-N concentration in the water and soil was minimal. $T_1$ reported the highest grain yield, total N content and uptake, whereas $T_7$ had the lowest values. On $T_1$, the apparent N recovery reached its highest level (73.5%). The NUE varied from 13.26% in $T_3$ to 29.38% in $T_1$. Based on this finding, deep placement of urea briquette by hand is recommended for increasing the yield and NUE of T. Aman rice.

**Keywords:** urea deep placement technology; $NH_4^+$-N concentrations; N use efficiency; apparent N recovery; T. Aman rice; grain yield



## 1. Introduction

In Bangladesh, nearly all habitats are lacking in nitrogen (N), which is the nutritional element most crucial to plant growth. Because of variations in weather conditions, methods of application, time of application, management practices, type of crop, variety, and so on, N use efficiency (NUE) varies from place to place and ecosystem to ecosystem [1–4]. The most widely applied form of N in Bangladesh is urea, but its efficiency is less than 50% in most crops [5]. Throughout the world, the NUE for rice in the study area is low, i.e., 30–35% of the applied urea, while the value is 33% for maize and 40–50% for potato [3]. High rates of N fertilizer input, incorrect timing of application, and broadcast application of N fertilizer are the main causes of poor NUE in the rice growing context [1]. The above-mentioned causes facilitate gaseous N loss pathways such as volatilization and denitrification, which correspond to about 16–20% and 0.75–1.0% of applied N, respectively, at the research location [2,6]. The management and timing of N fertilizer application are crucial for enhancing NUE in rice.

The N fertilizer plays a vital role in irrigated rice (*Oryza sativa* L.) cultivation. Despite the fact that even with the finest agronomic methods, only 30 to 40% of applied N in the form of urea is actually absorbed by the crop, current high irrigated rice yields are

typically connected with larger N-fertilizer doses. Due to numerous losses (volatilization, de-nitrification, runoff, seepage, and leaching), using nitrogenous fertilizer effectively is difficult. Increasing the effectiveness of assimilated N for grain yield has long been a challenge. The broadcast method is commonly used by most farmers in Bangladesh for applying urea in the rice field. Nutrients cannot be fully utilized by plant roots through broadcasting since they move laterally over a long distance. According to Iqbal [7], N losses from various N treatments during paddy growth varied between 2.82 and 5.07% when urea was applied in comparison to urea super granule (USG). The inadequate level of N recovery by rice plants is often due to high emissions of the soil–water–plant complex. Given the diminishing amount of farmland available, increasing N use efficiency (NUE) is essential for rice production in order to fulfill the rising demand for food. To achieve this, we tried to find out an effective method of N fertilizer application.

Deep-placement fertilization has now emerged as an effective substitute for broadcast fertilization in numerous investigations [8–10]. According to Kapoor et al. [11], deep placement of urea briquettes and NPK briquettes produced ammonium N levels in flood water that were on average 10 times lower than those produced by broadcast application of N as urea. The efficiency of fertilizers can be reduced by up to 70% when prilled urea (PU) is used. According to earlier research, compared to broadcasting, deep insertion of urea briquettes in NPK could greatly boost NUE and grain production [12]. Mechanically deep-placing urea (UDP) fertilizer boosts grain production and nitrogen use efficiency significantly when compared to surface broadcasting. In comparison to 30–45% for broadcast application, UDP boosts NUE by up to 80% with the elimination of losses. Consequently, compared to broadcast PU, UDP elevates grain yields by up to 20% while reducing the demand for N fertilizer by 30–35% in the cultivation of rice. Along with improving farm profitability, UDP reduces government subsidies in countries that provide N fertilizer subsidies.

In order to deliver plant nutrients to the crops, deep placement of urea refers to burying the N-fertilizer at a desired depth below the soil's surface using any equipment. It is an efficient way to arrest volatilization loss in a low land paddy cropping system. The Bangladesh Rice Research Institute (BRRI) has created a push-type USG applicator, which has a significant influence on raising nitrogen use efficiency. Additionally, it takes less time and effort than broadcasting. The International Fertilizer Development Corporation (IFDC) has been attempting to popularize the USG technique in several nations for a long time, but it has not been successful in Bangladesh due to a lack of a suitable applicator. Farmers must therefore laboriously and time-consumingly apply it by hand. It takes 200–300 h to apply urea to one hectare of land by hand [13]. Previous research showed that USG deep placement with a low range of N rate can reduce the need for urea fertilizer by up to 65% with a mean of 33%, and boost grain yields by up to 50% with an average boost of 15% to 20%, when compared to the split-prilled urea (PU) of the same amount [14]. Islam et al. [15] observed that the application of NPK briquette resulted in the maximum rice grain yield of 7.47 t ha$^{-1}$; and secondly, demonstrated superior agronomic efficiency than PU and USG. Additionally, they discovered that compared to recommended PU, NPK briquettes can save 33 kg of N per hectare.

The appropriate distribution of fertilizer at 7 to 10 cm depth with minimal nutrient leakage into surface soil or floodwater is necessary for the enhanced grain production, diminished N losses, and increased N usage efficiency [11]. Additionally, deep placement spacing needs to be constant to guarantee uniform growth and larger yields. Thus, the situation justifies the need for more research to be executed to assess the impacts of various UDP methods on the yield attributes and yield of rice under continuous flooding conditions. It also necessary to look into the effects of these methods on the N use efficiency of T. Aman rice.

## 2. Materials and Methods

### 2.1. Site and Soil Characteristics for the Experiment

The experimental site belongs to the non-calcareous dark grey floodplain soil under agro-ecological zone-17 (Old Brahmaputra Floodplain), and is situated at 24.75° N latitude and 90.5° E longitude with an 18 m elevation from sea level [16]. The Sonatala series of soil makes up the soil on the test field. The experiment was carried out in a typical silt loam soil during the 2015 Aman season at the Soil Science Field Laboratory, Bangladesh Agricultural University, Mymensingh. The soil had a silt loam texture with particle size distribution as follows: 3.64% sand, 78.18% silt, and 18.18% clay. The soil was relatively well-drained, with a pH of 6.27 (1:2.5 soil-water by glass electrode pH meter method [17]), 1.95% organic matter (Wet Oxidation method [18]), 0.14% total nitrogen (Kjeldahl method [19]), 3.16 ppm of available phosphorus (Olsen method [20]), 35.19 ppm of available potassium (Flame Photometer method after extraction with 1N $NH_4OAc$ at pH 7.0 [21], 348 $\mu S\ cm^{-1}$ electrical conductivity (Conductivity meter method using 1:5 soil-water extract), and 10.5 ppm of accessible sulfur (by extracting soil samples with 0.15% $CaCl_2$ and by measuring turbidity with Spectrophotometer [22]). The experimental area's climate was characterized by reasonably low temperatures during the Aman season and enough rainfall.

### 2.2. Crop and Variety

Rice was the test crop used in the experiment and the variety was BRRI dhan46, which is modern with a high yield. The Bangladesh Rice Research Institute is the organization that developed this variety. The variety is a transplanted Aman (T. Aman) rice variety that usually matures after 150 days after sowing with a mean yield of 4.7 t $ha^{-1}$ and receives 52 kg of nitrogen per ha from urea briquettes. The cultivar is a late non-lodging variant.

### 2.3. Treatments and Experimental Setup

The experiment was set up with a randomized complete block design (RCBD), in which the experimental area was divided into three blocks that represented the replications in order to curtail the heterogenic effects of the soil. There were seven different treatment combinations with three replications, which included: $T_1$: deep placement of urea briquette by hand; $T_2$: deep placement of urea briquettes by battery-powered applicator; $T_3$: deep placement of prilled urea briquettes by BRRI applicator; $T_4$: deep placement of urea briquettes by injector applicator; $T_5$: deep placement of urea briquettes by push-type applicator; $T_6$: broadcast application of prilled urea; and $T_7$: zero-N (control). The treatments were assigned at random to the seven unit plots that made up each block's subdivision. Consequently, there were twenty-one unit plots in total. Each plot was 6 m × 4 m in size, and they were separated from one another by an ail (30 cm). A 1-m-long drain separated the blocks from one another.

### 2.4. Agronomic Practices and Crop Management

With the use of a power tiller, the land was first prepped by ploughing and cross-ploughing, with laddering occurring at strategic intervals afterward. The land was subsequently cleaned, and after pondering, the plots were created in accordance with the idea by constructing an ail around each plot. In accordance with the treatment, the necessary urea fertilizers were applied. Water was removed from the rice plots prior to the application of N fertilizer. All the treatments except $T_6$ and $T_7$ received 52 kg N $ha^{-1}$ from urea briquettes (UB). Treatment $T_6$ (broadcast application of prilled urea) received 78 kg N $ha^{-1}$, and in $T_7$ (control), no N-fertilizer was applied. With the exception of the control treatment, the lowest N rate began at the then-currently-recommended rates of 52 kg N $ha^{-1}$ for deep placement and 78 kg N $ha^{-1}$ for broadcast application [4]. The size and number of briquettes for each placement site were determined based on N rates. One 2.7 g UB was deeply placed with a 40 cm × 40 cm interval.

Rice seedlings that were thirty days old were planted into the plots. There were three seedlings per hill, spaced 20 cm apart. To maintain the crop's typical growth and

development, intercultural operations such as weeding and pest control were carried out. Water was drained from the rice plots before N fertilizers were applied. Water was not permitted to linger on the field during the ripening phase. When the crop was fully mature, it was subsequently harvested. A 4 m$^2$ patch was taken from each plot and then threshed plot by plot.

### 2.5. Recording of Yield Components and Yields

At full crop maturity, the BRRI dhan46 growth parameters (plant height, effective tillers per hill, panicle length, and grains per panicle) were measured. The yields of various treatments were also measured after harvest. Five hills, or panicles, on average, were present in each observation. The harvested samples from each treatment were used to determine the yield attributes (1000-grain weight, grain, and straw yields) of BRRI dhan46, and the weight was measured after oven drying. The straw yield was calculated using a sundry basis, whereas the grain yield was calculated using a 14% moisture basis.

### 2.6. Determination of Field Water $NH_4^+$-N and $NO_3^-$-N

For the aim of determining $NH_4^+$-N and $NO_3^-$-N, surface water samples from the rice field were taken three times, i.e., after each split urea application. Seven days following every fertilizer treatment, samples of floodwater were taken. To measure $NH_4^+$-N and $NO_3^-$-N, the obtained water samples were taken to the lab. Every day, the average water depth was measured by scaling each plot. Irrigation was applied when required to maintain the desired water depth (6 cm) in the field. The pH of the surface water was measured as soon as possible after sampling at the Soil Science Laboratory by a glass electrode pH meter.

Using a spectrophotometer at a wavelength of 640 nm, floodwater samples (25 mL water +2.0 mL phenol solution, 2.0 mL Na-nitroprusside, and 5.0 mL oxidizing solutions) were analyzed for $NH_4^+$-N [23]. For the determination of $NO_3^-$-N in rice field water, 25 mL of water sample plus 0.8 mL of 5% salicylic-$H_2SO_4$ solution were boiled and allowed to cool at room temperature. After cooling, 19 mL of 2N NaOH solution was added to raise the pH to more than 12. Following that, solution absorbance was determined using a spectrophotometer at 410 nm [24]. By creating a standard curve using a standard solution, the value of $NH_4^+$-N and $NO_3^-$-N was calculated from absorbance and expressed in ppm. In order to correctly account for the area and volume of water per plot, the concentrations of $NH_4^+$-N and $NO_3^-$-N (ppm or mg L$^{-1}$) were converted to kg ha$^{-1}$.

### 2.7. Determination of Soil $NH_4^+$-N and $NO_3^-$-N

At a depth of 2 cm beneath the surface, the initial soil samples were taken. These samples were taken at 1 through 5 days and the 7th and 10th days after fertilizer application. For the deep-placement treatments ($T_1$ to $T_5$), the samples were collected from the rows receiving the fertilizer. A quadrant of 10 cm long, 5 cm wide, and 2 cm deep was placed as part of the sampling. This procedure was repeated for three locations in the fertilizer row to make a composite sample. The samples were then stored in a clean ice-cooler and immediately analyzed within 5 days. All samples were taken from a designated area of each plot. Standard procedures were used to evaluate the initial soil samples for chemical elements including $NO_3^-$-N and $NH_4^+$-N [25]. The extracts of nitrate and ammonium values from soil contents were analyzed utilizing continuous flow colorimetry. Post-analysis calculation of soil contents from the extract's nitrate and ammonium values were calculated as follows:

μg analyte per g dry soil = (measured value, mg L$^{-1}$) × 1000 μg mg$^{-1}$ × 0.02 L/g dry soil (g)

[The dry weight value of the soil sample is based on percent moisture data for each sample, obtained separately].

*2.8. Plant Sample Analysis for N Determination*

By using the micro-Kjeldahl method, total N was calculated from 1 g of oven-dry powdered material, 1.1 g of catalyst mixture ($K_2SO_4$: $CuSO_4.5H_2O$: Se = 100:10:1), 2 mL 30% $H_2O_2$, and 3 mL $H_2SO_4$ [19]. The total N concentration was measured using the following equation [26]:

$$\text{Total N (\%)} = \frac{(\text{Titration value of sample} - \text{Titration value of blank}) \times \text{ Conc. of } H_2SO_4 \times 0.014 \times \text{ dillution factor } \times 100}{\text{Soil sample weight}}$$

*2.9. Nutrient Uptake*

Nutrient uptake by grain and straw was calculated by the following equation [27–29]:

$$\text{Nutrient uptake (kg/ha)} = \frac{\text{Grain yield } \times \text{Grain N content} + \text{Straw yield } \times \text{Straw N content}}{100}$$

*2.10. Measurement of ANR*

The kilogram of nitrogen absorbed per kilogram of fertilizer applied is known as apparent nitrogen recovery. To calculate ANR the following equation was employed [30]:

$$\text{ANR (\%)} = \frac{\text{Total N uptake in a specific treatment } - \text{Total N uptake in control}}{\text{Fertilizer N applied}} \times 100$$

*2.11. Determination of NUE*

An increase in grain production per kilogram of applied nitrogen is what is meant by "nitrogen utilization efficiency." Using various applicators and dosages, N fertilizers were applied on several plots. NUE was calculated for each treatment and accounts for agronomic efficiency, kg N absorption per kg N supplied, and kg grain per kg N applied. The NUE was determined using the following equation [31]:

$$\text{NUE} = \frac{\text{Grain yield in a specific treatment } - \text{ Grain yield in control}}{\text{Fertilizer N applied (kg/ha)}}$$

*2.12. Statistical Analysis*

A complete randomized design served to measure the analyses of variance (ANOVA) for various crop characteristics, yields, N concentration and uptake, ANR, and NUE. Differences between the treatment means were determined by Duncan's multiple range test (DMRT) at a 5% level of probability [32] using Minitab 17 software.

## 3. Results

### 3.1. $NH_4^+$-N and $NO_3^-$-N Concentration in Field Water

To track the release of $NH_4^+$-N from nitrogenous fertilizers, the concentration of ammonium in the surface water of the experimental field was measured every day for seven days after the application of USG, briquettes, and prilled urea applied by various applicators. Figure 1 depicts the trends of $NH_4^+$-N release pattern in water samples as influenced by various treatments. After 1 day of application, the amount of $NH_4^+$-N in rice field water released from prilled urea by broadcasting and by BRRI applicators ($T_6$ and $T_3$) began to rise, lasted for 4 days, and then progressively dropped for 7 days (Figure 1). Surface water $NH_4^+$-N levels did not significantly differ amongst the treatments ($T_1$, $T_2$, $T_4$, $T_5$, and $T_7$). However, during the sample time, a continual release of $NH_4^+$-N was seen in the rice field water. The highest ammonium concentration in water (16.23 ppm) was found in $T_6$ treatments. The nitrate concentration in the water sample was negligible and near about zero.

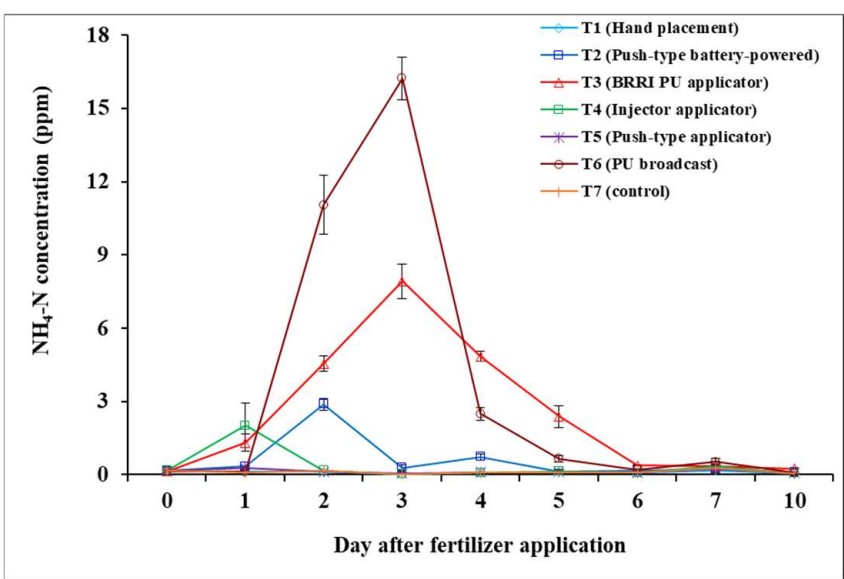

**Figure 1.** Trend of $NH_4^+$-N release in flood water under different methods of urea fertilizer application in T. Aman rice field at BAU Farm.

### 3.2. $NH_4^+$-N and $NO_3^-$-N Concentration in the Soil

Figure 2 shows the trend of $NH_4^+$-N release patterns in soil samples under various treatments. The amount of available $NH_4^+$-N in soil released from $T_3$ started to increase after 1 day of application and reached its maximum after 5 days of application and then gradually decreased. When fertilizer was applied using a push-type applicator ($T_5$), the amount of $NH_4^+$-N released rose after 5 days, then fell after 7 and 10 days. Following fertilizer application for 7 days in $T_1$ and $T_7$, the amount of $NH_4^+$-N released increased, and subsequently it dropped, after fertilizer application for 10 days. Similarly, to field water, $NO_3^-$-N concentration was also negligible in soil.

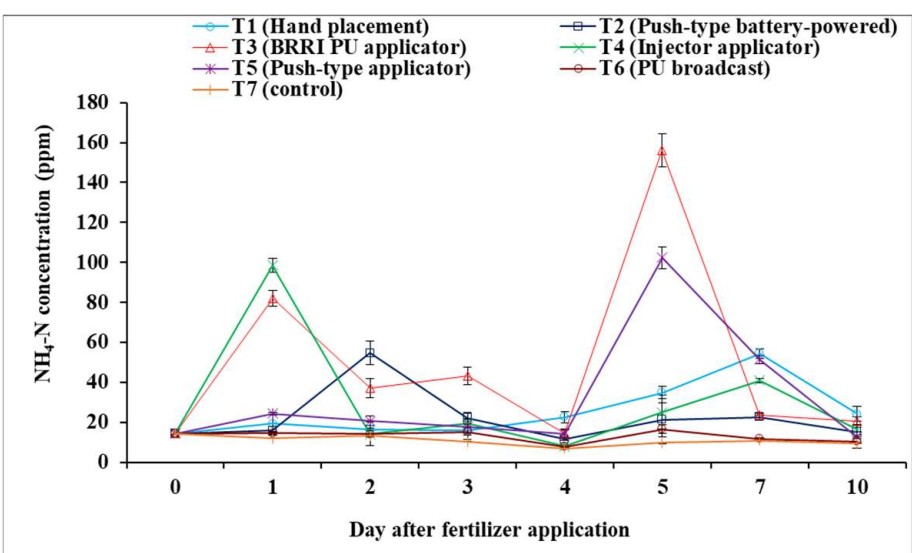

**Figure 2.** Trend of soil $NH_4^+$-N release under different methods of urea fertilizer application in T. Aman rice field at BAU Farm.

### 3.3. Yield Attributes of BRRI dhan46

Table 1 shows the response of different yield attributes of BRRI dhan46 to different methods of deep placement of urea application. Although no appreciable variations in the BRRI dhan46 plants' heights were evident as a result of using various urea application

techniques, the plant heights of various treatments were all significantly higher than $T_7$ (control). $T_4$ had the most effective tillers per hill (10.90), and it statistically resembled $T_1$, $T_2$, $T_3$, and $T_6$, respectively. $T_7$ treatments revealed the least effective tillers $hill^{-1}$ value of 6.20.

**Table 1.** Effect of methods of deep placement of urea fertilizer on the yield components of BRRI dhan46.

| Treatment | Plant Height (cm) | Effective Tillers Hill$^{-1}$ (No.) | Panicle Length (cm) | Filled Grains Panicle$^{-1}$ (No.) | 1000-Grain Weight (g) |
|---|---|---|---|---|---|
| $T_1$ | 96.77 a | 10.73 a | 21.79 a | 168.60 ab | 26.73 ns |
| $T_2$ | 95.00 a | 9.67 ab | 22.42 a | 171.80 ab | 26.20 ns |
| $T_3$ | 92.97 a | 9.23 ab | 21.77 a | 161.47 ab | 26.43 ns |
| $T_4$ | 97.00 a | 10.90 a | 22.06 a | 163.97 ab | 25.82 ns |
| $T_5$ | 95.00 a | 8.67 b | 22.50 a | 180.70 a | 26.25 ns |
| $T_6$ | 92.90 a | 10.10 ab | 21.41 a | 150.33 b | 26.87 ns |
| $T_7$ | 81.30 b | 6.20 c | 18.86 b | 103.63 c | 25.82 ns |
| CV % | 3.50 | 10.10 | 3.12 | 8.40 | 1.75 |
| SE ($\pm$) | 1.883 | 0.546 | 0.389 | 7.626 | 0.154 |

Note(s): Figures in a column having common letters do not differ significantly at 5% level of significance. CV (%) = coefficient of variation; SE ($\pm$) = standard error of means; ns = not significant; $T_1$: deep placement of urea briquettes by hand; $T_2$: deep placement of urea briquettes by battery-powered applicator; $T_3$: deep placement of prilled urea briquettes by BRRI applicator; $T_4$: deep placement of urea briquettes by injector applicator; $T_5$: deep placement of urea briquettes by push-type applicator; $T_6$: broadcast application of prilled urea; $T_7$: zero-N; control.

All the methods of deep placement gave significantly higher panicle lengths than the control ($T_7$). However, the panicle length produced by different treatments except $T_7$ was statistically similar, although there were numerical variations among the panicle lengths. When applying several applicators for deep urea placement, the filled grains panicle$^{-1}$ of BRRI dhan46 outperformed the control $T_7$. It is noted that the effects of the treatments were insignificant, despite the fact that all of them led to larger 1000-grain weights than the control.

### 3.4. Grain and Straw Yield

As demonstrated in Table 2, the various urea deep placement approaches considerably boosted the grain and straw yields of BRRI dhan46. Despite significant quantitative variations in grain production between the treatments, $T_7$, where no N fertilizer was used, reported the lowest value (3085 kg ha$^{-1}$). The grain yield of $T_1$ was the highest (4613 kg ha$^{-1}$), and it was statistically comparable to that of $T_2$, $T_4$, $T_5$, and $T_6$. Based on the increase in grain yield over control, the treatments were assessed in the following order: $T_1 > T_4 > T_5 > T_2 > T_6 > T_3 > T_7$. The treatments were graded in order of straw yield as follows: $T_5 > T_2 > T_1 > T_6 > T_4 > T_3 > T_7$. As shown in Table 2, $T_5$ recorded the biggest increase (91.64%) while $T_3$ recorded the smallest one (49.0%) in terms of the percentage increase in straw yield compared to control.

### 3.5. Nitrogen Content and Uptake

There were no appreciable differences in the nitrogen content of the grain and straw of BRRI dhan46 between the urea application techniques' treatments (Table 3). $T_1$ had the highest N content, at 1.1%, while $T_7$ had the lowest N content, at 1.01%. From 0.52% in $T_7$ to 0.62% in $T_1$, straw contained N. As a result of various urea application techniques, the findings show that rice grain possessed a larger N concentration than straw.

**Table 2.** Grain and straw yields of BRRI dhan46 as influenced by methods of deep placement of urea fertilizer.

| Treatments | Grain Yield (kg ha$^{-1}$) | % Increase over Control | Straw Yield (kg ha$^{-1}$) | % Increase over Control |
|---|---|---|---|---|
| $T_1$ | 4613 a | 49.51 | 5651 a | 81.05 |
| $T_2$ | 4496 a | 45.72 | 5784 a | 85.31 |
| $T_3$ | 3775 b | 22.34 | 4648 b | 49.00 |
| $T_4$ | 4602 a | 49.14 | 5441 a | 74.30 |
| $T_5$ | 4512 a | 46.24 | 5982 a | 91.64 |
| $T_6$ | 4142 ab | 34.26 | 5526 a | 77.03 |
| $T_7$ | 3085 c | - | 3121 b | - |
| CV % | 6.47 | - | 8.33 | - |
| SE ($\pm$) | 155.9 | - | 256.9 | - |

Note(s): Figures in a column having common letters do not differ significantly at 5% level of significance. CV (%) = coefficient of variation; SE ($\pm$) = standard error of means; $T_1$: deep placement of urea briquettes by hand; $T_2$: deep placement of urea briquettes by battery-powered applicator; $T_3$: deep placement of prilled urea briquettes by BRRI applicator; $T_4$: deep placement of urea briquettes by injector applicator; $T_5$: deep placement of urea briquettes by push-type applicator; $T_6$: broadcast application of prilled urea; $T_7$: zero-N; control.

**Table 3.** Effect of methods of deep placement of urea fertilizer on N content and uptake of BRRI dhan46.

| Treatments | N Content (%) | | N Uptake (kg ha$^{-1}$) | | |
|---|---|---|---|---|---|
| | Grain | Straw | Grain | Straw | Total |
| $T_1$ | 1.10 | 0.62 | 50.85 a | 34.83 a | 85.68 a |
| $T_2$ | 1.03 | 0.52 | 46.16 ab | 31.67 a | 77.87 ab |
| $T_3$ | 1.03 | 0.58 | 38.74 c | 33.99 a | 72.73 b |
| $T_4$ | 1.06 | 0.58 | 48.93 a | 31.45 a | 80.38 ab |
| $T_5$ | 1.06 | 0.56 | 47.94 a | 33.42 a | 81.36 ab |
| $T_6$ | 1.03 | 0.54 | 42.50 bc | 30.11 a | 72.61 b |
| $T_7$ | 1.01 | 0.52 | 31.11 d | 16.35 b | 47.46 c |
| CV % | 4.39 | 11.99 | 6.63 | 15.95 | 8.05 |
| SE ($\pm$) | 0.011 | 0.014 | 1.674 | 2.786 | 4.766 |

Note(s): Figures in a column having common letters do not differ significantly at 5% level of significance. CV (%) = coefficient of variation; SE ($\pm$) = standard error of means; $T_1$: deep placement of urea briquettes by hand; $T_2$: deep placement of urea briquettes by battery-powered applicator; $T_3$: deep placement of prilled urea briquettes by BRRI applicator; $T_4$: deep placement of urea briquettes by injector applicator; $T_5$: deep placement of urea briquettes by push-type applicator; $T_6$: broadcast application of prilled urea; $T_7$: zero-N; control.

Due to various urea application techniques, the product and byproduct of BRRI dhan46 dramatically boosted their N uptake. The lowest N uptake (47.46 kg ha$^{-1}$) was identified in $T_7$, whereas the maximum N uptake (85.68 kg ha$^{-1}$) was also noted in $T_1$, which was statistically comparable to $T_2$, $T_4$, and $T_5$. Regarding the N uptake, the treatments were ranked as $T_1 > T_5 > T_4 > T_2 > T_3 > T_6 > T_7$.

*3.6. Apparent Nitrogen Recovery (ANR) and Nitrogen Use Efficiency (NUE)*

Table 4 shows the ANR and NUE values of BRRI dhan46 as influenced by various urea application techniques. In various treatments, the mean ANR for rice ranged from 32.2 to 73.5%. $T_1$ produced the highest ANR (73.5%), which was followed by $T_5$ (65.2%), $T_4$ (63.3%), $T_2$ (58.4%), and $T_3$ (48.6%). $T_6$ produced the lowest ANR (32.2%). $T_1$ had the highest NUE value (29.38 kg grain yields rise for each kg of added nitrogen), followed by $T_4$ (29.16 kg grain yields rise for each kg of added nitrogen), $T_5$ (27.44 kg grain yields rise for each kg of added nitrogen), and $T_2$ (27.13 kg grain yields rise for each kg of added nitrogen). The lowest value (13.26) was observed in the $T_3$ treatment, whereas the NUE (13.55 kg grain increase from PU) was attained in $T_6$.

**Table 4.** Effects of different methods of deep placement of urea fertilizer on apparent N recovery (ANR) and N use efficiency (NUE) of BRRI dhan46.

| Treatments | N Applied (kg ha$^{-1}$) | NUE | ANR (%) |
|:---:|:---:|:---:|:---:|
| $T_1$ | 52 | 29.38 | 73.5 |
| $T_2$ | 52 | 27.13 | 58.4 |
| $T_3$ | 52 | 13.26 | 48.6 |
| $T_4$ | 52 | 29.16 | 63.3 |
| $T_5$ | 52 | 27.44 | 65.2 |
| $T_6$ | 78 | 13.55 | 32.2 |
| $T_7$ | - | - | - |

Note(s): NUE = N use efficiency (NUE), % ANR = apparent N recovery; $T_1$: deep placement of urea briquettes by hand; $T_2$: deep placement of urea briquettes by battery-powered applicator; $T_3$: deep placement of prilled urea briquettes by BRRI applicator; $T_4$: deep placement of urea briquettes by injector applicator; $T_5$: deep placement of urea briquettes by push-type applicator; $T_6$: broadcast application of prilled urea; $T_7$: zero-N; control.

## 4. Discussion

Numerous studies have reported that slow-released N fertilizer is an effective management strategy for improving NUE in rice [9,33,34]. When prilled urea is used in flooded conditions, it quickly hydrolyzes and releases more $NH_4^+$-N in the soil and water over the course of a few days, which promotes N loss through denitrification and volatilization. In contrast, use of USG causes slow release of N, which helps uniform plant uptake for a longer period of time by supplying available N. Our findings showed that following the prilled urea broadcast application and the subsequent application by BRRI applicator, increased $NH_4^+$-N levels were found within 2–3 days. These findings concur with those of Jahan et al. [35], who discovered the highest concentration of ammonium nitrogen on day 2 following the application of prilled urea in rice field water. According to comparable findings by Liu et al. [8] and Uddin et al. [2], the abrupt decrease in the concentration of floodwater ammonium after 5 to 6 days of broadcasting prilled urea can be attributed to gaseous N losses, $NH_4^+$-N dispersion into the soil, and nitrification process. Similarly, soil $NH_4^+$-N concentrations varied from method to method and reached their peak values up to 7 days and subsequently declined decreased thereby. This is attributed to the slow release of N and vigorous urease activity in soil up to 7 days after urea application [2]. The $NO_3^-$-N production in soil was negligible compared to the $NH_4^+$-N concentration in soil and water. This is due to the presence of standing water in the field, which obstructed the presence of oxygen for the nitrification process [26]. For minimizing floodwater ammonium-N and ammonia volatilization loss, placement of any N material with simultaneous soil cover is an effective management approach [36].

Our findings demonstrated that methods of N fertilization considerably improved the yield characteristics of T. Aman rice. These results concur with those of Saiful et al. [37], who discovered that the application method for urea had no appreciable impact on panicle intensity, panicle length, or 1000-grain mass. The results showed that the impacts of various nitrogenous fertilizer application techniques on the growth characteristics of BRRI dhan46 were comparable to those of UDP on the yield and NUE of BRRI dhan46 under flooded conditions, which were also previously discovered by Islam et al. [38]. Urea briquettes placed deep into the soil improved N uptake, and more N assimilation led to more productive tillers and full grains, which ultimately increased the T. Aman rice yield. When compared to other applicators, which were closely coordinated with Saiful et al. [37], deep placement of urea briquettes by hand performed better in boosting the grain yield of rice. Nitrogen application methods also significantly influenced the N uptake, apparent recovery, and NUE. All these parameters were almost double in deep placement of urea briquettes by either hand or any other applicator than those of prilled urea. This is due to the slow release of N briquettes, which resulted in less ammonium in the soil-water system for volatilization as well as denitrification losses, and higher N intake and assimilation due to prolonged nutrient supply in the plant-water system. Therefore, deep placement of urea briquettes by hand or any other applicator is the most effective method for improving NUE.

## 5. Conclusions

In recent years, urea deep placement (UDP) using various techniques has emerged as one of the most effective managerial strategies for increasing crop yield and improving the NUE of T. Aman rice. Overall, the findings proved that UDP by hand ($T_1$) performed better than the other deep placement methods regarding crop yields, ANR, and NUE. Therefore, for the successful growing of transplanted Aman rice, urea briquette application by hand is advised. The advantage of UDP by hand is that this method makes feasible perfect placement of urea briquettes. It is, however, more time-consuming and labor-extensive. Additional investigation is required to calculate the amount of N lost from flooded rice fields under these treatments in the forms of $NH_3$, $NO_3$, and NOx ($N_2O$ and NO), in order to draw a generalized conclusion.

**Author Contributions:** Conceptualization, A.A., M.R.I. (Mohammad Rafiqul Islam) and M.R.I. (M. Rafiqul Islam); methodology, A.A., M.R.I. (Mohammad Rafiqul Islam) and M.R.I. (M. Rafiqul Islam); software, A.A., M.R.I. (Mohammad Rafiqul Islam) and M.R.I. (M. Rafiqul Islam); validation, M.R.I. (Mohammad Rafiqul Islam) and M.R.I. (M. Rafiqul Islam); formal analysis, A.A., M.R.I. (Mohammad Rafiqul Islam), M.A.I., S.L.H. and S.U.; investigation, M.R.I. (Mohammad Rafiqul Islam) and M.R.I. (M. Rafiqul Islam); resources, A.A., M.R.I. (Mohammad Rafiqul Islam) and M.R.I. (M. Rafiqul Islam); data curation, A.A., M.R.I. (Mohammad Rafiqul Islam), M.R.I. (M. Rafiqul Islam) and S.U.; writing—original draft preparation, A.A., M.R.I. (Mohammad Rafiqul Islam), M.A.I. and S.U.; writing—review and editing, A.A., M.R.I. (Mohammad Rafiqul Islam), M.R.I. (M. Rafiqul Islam), M.A.I., S.L.H., S.U. and M.M.R.; visualization, M.R.I. (Mohammad Rafiqul Islam) and M.R.I. (M. Rafiqul Islam); supervision, M.R.I. (Mohammad Rafiqul Islam) and M.R.I. (M. Rafiqul Islam); project administration, M.R.I. (Mohammad Rafiqul Islam) and M.R.I. (M. Rafiqul Islam); funding acquisition, M.R.I. (Mohammad Rafiqul Islam), M.R.I. (M. Rafiqul Islam) and M.M.R. All authors have read and agreed to the published version of the manuscript.

**Funding:** The United States Agency for International Development (USAID) provided support for this research through the project "Accelerating Agriculture Productivity Improvement-Integrating Greenhouse Gas Emissions Mitigation into the Feed the Future Bangladesh Fertilizer Deep Placement Rice Intensification (Cooperative Agreement Number AID-388-A-10-00002)".

**Data Availability Statement:** The data that support this study will be shared upon reasonable re-quest to the corresponding author.

**Acknowledgments:** The authors offer gratitude to USAID- for their support to complete this research.

**Conflicts of Interest:** The authors declare no conflict of interest.

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
