# Peer review of "Methods of Urea Fertilizer Application Influence Growth, Yield, and Nitrogen Use Efficiency of Transplanted Aman Rice"

_water, doi:10.3390/w14213539_

Round 1
Reviewer 1 Report
This study explored the effects of different nitrogen application techniques on the nitrogen use efficiency of rice, which has certain guiding significance for improving the fertilization pattern, but there are also some problems, which are marked as "highlighted" in the PDF file.

Author Response
Comment: This study explored the effects of different nitrogen application techniques on the nitrogen use efficiency of rice, which has certain guiding significance for improving the fertilization pattern, but there are also some problems, which are marked as "highlighted" in the PDF file.
Response: Thank you very much for your comments.
Comment: Rice nitrogen mainly comes from nitrogen fertilizer? Fill in the references.
Response: There are several sources of rice nitrogen. But urea is the most widely applied form of nitrogen in the rice field of Bangladesh. Reference has been added in line 41.
Comment:
Focus on the questions explored in this study and do not mention directions that are not covered.
Response: Thank you very much for your suggestion. The questions explored in this study have been focused. Please see line 65.
Comment:
Lines 332-343, these should not continue to appear here, but in the introduction. These are the reasons why you did this research, not the “discussion”.
Response: That portion has been removed from ‘discussion’ and transferred to the ‘introduction’.
Comment:
Although “T1” in this study has a good effect, it is more time-consuming and labor-expensive, so what are its advantages? What are the implications for practical field operations? Further investigation did not solve these problems.
Response: We are agreed with your comments. The advantage of UDP by hand (T1) is that perfect placement of urea briquette is possible by this method which has been mentioned in lines 430-431. Although, it is time consuming and labor expensive, it is being used in field operation by the farmers.
Reviewer 2 Report
The Revised manuscript entitled "Deep Placement Methods of Urea Fertilizer Application Influence Flood Water Nitrogen, Yield and Nitrogen Use Efficiency of Transplanted Aman Rice" describes the results on how different urea application techniques affected the yield and nitrogen use efficiency (NUE) of transplanted Aman rice (cv. BRRI dhan46).
Due to nitrogen fertilization problems, the results presented in revised manuscript results are important. The manuscript is well-prepared, easy to follow, and interesting to readers. However, before final publication, it requires a few minor corrections.
I suggest providing more literature position to the Introduction section. This section is based only on seven literature positions that need to be more.
Subsection 2.1
Line 91 - Sentence: "Sand made up....please put after sentence...The soil had a silt loam texture witch particle size distribution as follows: 3.64% Sand, 78.18 silt, and 18.18% clay...(line 88).
Line 90 -avilabile phosphorus - which method was utilized,
exchangeable potasium - which method was used. I suggest providing available potasium in ppm like phosphorus. Exchangable cations should be presented together with Ca2+, Mg2+, etc., as complex sorption characteristics.
Table 1, last column - there are no letters describing the significance of differences for individual values.
My recommendations - minor remarks
Author Response
Comment: The Revised manuscript entitled "Deep Placement Methods of Urea Fertilizer Application Influence Flood Water Nitrogen, Yield and Nitrogen Use Efficiency of Transplanted Aman Rice" describes the results on how different urea application techniques affected the yield and nitrogen use efficiency (NUE) of transplanted Aman rice (cv. BRRI dhan46). Due to nitrogen fertilization problems, the results presented in revised manuscript results are important. The manuscript is well-prepared, easy to follow, and interesting to readers. However, before final publication, it requires a few minor corrections.
Response: Thank you very much for your comments.
Comment: I suggest providing more literature position to the Introduction section. This section is based only on seven literature positions that need to be more.
Response: More literature positions have been provided in the introduction. Now it contains 16 literature positions. Please see lines 36-105.
Comment: Subsection 2.1 Line 91 - Sentence: "Sand made up.... please put after sentence...The soil had a silt loam texture witch particle size distribution as follows: 3.64% Sand, 78.18 silt, and 18.18% clay... (line 88).
Response: The sentence has been revised accordingly. Please see lines 114-115.
Comment: Line 90 -avilabile phosphorus - which method was utilized, exchangeable potasium - which method was used. I suggest providing available potasium in ppm like phosphorus. Exchangable cations should be presented together with Ca2+, Mg2+, etc., as complex sorption characteristics.
Response: The methods of determining available phosphorus and potassium have been included in lines 118-120. We have also added the methods of measuring other soil parameters. Please see lines 116-118 and 121-123. The available potassium has been expressed in ppm. It would be fine to present exchangeable cations together with Ca2+, Mg2+, etc., as complex sorption characteristics but we are sorry that those were not determined.
Comment: Table 1, last column - there are no letters describing the significance of differences for individual values.
Response: In last column of Table 1, ‘ns’ has been mentioned which means ‘not significant’ as indicated in the footnote.